# Easy Synthesis and Characterization of Novel Carbon Dots Using the One-Pot Green Method for Cancer Therapy

**DOI:** 10.3390/pharmaceutics14112423

**Published:** 2022-11-09

**Authors:** Lijie Wang, Donghao Gu, Yupei Su, Dongxu Ji, Yue Yang, Kai Chen, Hao Pan, Weisan Pan

**Affiliations:** 1School of Pharmacy, Shenyang Medical College, Shenyang 110034, China; 2School of Pharmacy, Shenyang Pharmaceutical University, Shenyang 110016, China; 3School of Pharmacy, Liaoning University, Shenyang 110036, China

**Keywords:** hyaluronic acid, carboxymethyl chitosan, carbon dots, bio-imaging, antitumor drug delivery

## Abstract

In this study, hyaluronic acid (HA) and carboxymethyl chitosan (CMCS) were used for the synthesis of novel targeted nanocarrier carbon dots (CD_C-H_) with photo-luminescence using a one-step hydrothermal method. Doxorubicin (DOX), a common chemotherapeutic agent, was loaded with the CD_C-H_ through electrostatic interactions to form DOX–CD_C-H_ complexes as a targeted antitumor drug delivery system. The synthesized CD_C-H_ show a particle size of approximately 6 nm and a high fluorescence quantum yield of 11.64%. The physical and chemical character properties of CD_C-H_ and DOX–CD_C-H_ complexes were investigated using various techniques. The results show that CD_C-H_ have stable luminescent properties and exhibit excellent water solubility. The in vitro release study showed that DOX–CD_C-H_ exhibited pH-dependent release for 24 h. Confocal laser scanning microscopy was applied to investigate the potential of CD_C-H_ for cell imaging and the cellular uptake of DOX–CD_C-H_ in different cells (NIH-3T3 and 4T1 cells), and the results confirmed the target cell imaging and cellular uptake of DOX–CD_C-H_ by specifically binding the CD_44_ receptors on the surface of tumor cells. The r MTT results suggest that the DOX–CD_C-H_ complex may induce apoptosis in 4T1 cells, reducing the cytotoxicity of free DOX-induced apoptosis. In vivo antitumor experiments of DOX–CD_C-H_ exhibited enhanced tumor cancer therapy. CD_C-H_ have potential applications in bioimaging and antitumor drug delivery.

## 1. Introduction

In recent years, the family of carbon nanomaterials consisting of carbon nanofibers, graphene, nanotubes, fullerenes, nanodiamonds and carbon dots has been particularly active in nanotechnology research. Among these, carbon dots (CDs) are a recently discovered member with excellent properties and are currently recognized as a functional nanomaterial. Their unique characteristics such as facile synthesis, high aqueous solubility, low environmental toxicity, strong and tunable photo-luminescence, easy functionalization, excellent biological and compatibility have attracted tremendous attention in various fields [1,2]. Olga B et al. produced time-stimulus responsive green fluorescent nanoparticles by combining silica nanoparticles (SNs) and green emitting carbon dots (CDs) to detect glutathione with a lower limit of detection of 0.15 µM [3]. Aparajita G et al. reported the folic acid (FA)-functionalized fluorescent CDs loaded with glucose oxidase (GOx) and anticancer agent paclitaxel (PTX) (FA−CD−(PTX−GOx)) were screened for their anti-TNBC cell potential and three-dimensional breast tumor spherical model [4]. Meizhe Y et al. reported folate-derived carbon dots (FA-CDs) that act as a simple and effective antimicrobial nanosystem that can respond to environmental stimuli of bacterial infection and acquire antimicrobial capacity “on demand” [5]. Researchers have made use of carbon dots as alternatives to novel nano-nontoxicity materials in the medical fields of biosensors, photoluminescence, gene drug delivery, biological probes and bioimaging [6,7,8]. In general, there are two main methods for the synthesis of carbon dots: top-down and bottom-up strategies [9,10]. Top-down is a method to decompose large particles into smaller nanometer-sized ones via physical and chemical methods such as laser etching, arc discharge, electrochemical synthesis and plasma treatment. For bottom-up methods (e.g., hydrothermal treatment, oxidation method, microwave and ultrasonic assisted methods), CDs of desired size range are commonly formed by carbonization of small particles, involving carbohydrates and a passivator. The one-step hydrothermal method, a green synthesis strategy method with environmentally friendly solvents using simple equipment, has become the dominant method for the mass preparation of carbon dots. The carbon sources of synthetic CDs are extensive, including milk [11], tea [12], egg [13], honey [14], flowers [15], sodium citrate [16], albumin [17] and chitosan [18], etc. Biocompatibility and the pollution-free and environmentally friendly nature of CD’s using biomolecular materials as carbon sources are attracting a lot of attention. Dopant atoms, such as sulfur, nitrogen, etc., to improve fluorescence quantum yield CDs are increasingly selected as raw materials for carbon dots [19,20]. As a result, the materials with dopant atoms, superior biocompatible and excellent water solubility have become the preferred carbon sources for simple one-step hydrothermal methods.

Chitosan (CT) is a natural renewable cationic polymer with high biocompatibility, permeability promotion, safety and antibacterial activity, and it has been used as a drug carrier in the field of medicine [21,22]. When chitosan is used as a carbon source to prepare carbon dots by a simple one-step hydrothermal method, the functional groups of chitosan will remain on the surface of the CDS and result in good water solubility, stable luminescence performance, low cost and excellent biocompatibility of the synthesized carbon dots [18]. However, the poor water solubility of chitosan limits its application as a carbon source for CDs preparation. Carboxymethyl chitosan (CMCS) due to the introduction of carboxymethyl, which destroys the secondary structure of chitosan molecules and considerably reduces the crystallinity of CT, has become amorphous with better water solubility [23]. Thus, based on the research of chitosan CDs [24], CMCS was selected as a biodegradable and safe carbon source for the synthesis of biologically friendly N-doped carbon dots with a high fluorescence quantum yield.

Despite synthetic CD_S_ having shown numerous advantages in the field of medicine, side effects and harmful effects on normal tissues and cells are still a difficult problem that should be solved [25]. There are many functional groups on the surface of CD_S_ that can connect to the active targeted moieties and become the targeted specificity drug carrier to reduce toxic and side effects [26]. However, surface modification involves complicated reactions, multiple steps and low yields. Therefore, it is necessary to develop self-targeting CD_S_ for precision drug delivery systems.

A typical biofriendly material is the targeted molecular hyaluronic acid (HA), which has a strong affinity to CD_44_ receptors and which is overexpressed on various tumor cells [27]. HA, also known as Restylane, is a linear macromolecular acid mucopolysaccharide widely distributed in humans and animals. In the human body, it is found in the skin and connective tissue as an extracellular matrix. In addition to providing water and volume to cells, it is characterized by tissue stability, strong binding forces, high viscoelasticity, weak species and tissue diversity, and no immunogenicity. With excellent water solubility and biocompatibility, HA is widely used in the fields of drug delivery, cardiology, arthritis, cancer treatment and health food [28]. It has been reported that as a carbon source [29,30,31], HA was successfully added with a passivator to prepare CDs as a self-targeting CD_HA_, to deliver loaded drug release into tumor cells, and for bio-imaging. In this study, on the basis of CMCS as a carbon source, HA was directly added to the hydrothermal reaction to in situ synthesis of self-targeted carbon dots [32,33]. In this study, doxorubicin (DOX), a cationic antitumor drug, was selected as a model drug and loaded with CD_S_ by electrostatic interaction to form a targeted antitumor drug delivery system.

In this paper, novel green self-targeted carbon dots were synthesized by one-step hydrothermal carbonization of HA and CMCS for targeted antitumor drug delivery as illustrated in Figure 1. Various analytical experimental techniques were used to confirm the successful preparation of the CD_C-H_ and DOX–CD_C-H_ complexes. The in vitro release study showed that DOX–CD_C-H_ has pH-dependent DOX release behavior, and the imaging and cellular uptake of DOX–CD_C-H_ on NIH-3T3 and 4T1 cells were investigated using a laser scanning confocal microscope. In addition, MTT assays have shown that the DOX–CD_C-H_ complex can induce apoptosis in 4T1 cells, reducing the cytotoxicity of free DOX-induced apoptosis. Finally, a series of in vivo experiments were conducted to investigate the antitumor activity and biological safety of the novel targeted drug delivery system. The targeted nanocarriers presented in this paper may provide a promising candidate for cell imaging and antitumor drug delivery.

## 2. Materials and Methods

### 2.1. Materials

Doxorubicin hydrochloride (DOX·HCl) and ethylenediamine (EA) were purchased from Macklin^®^ (Shanghai, China). Carboxymethyl chitosan (CMCS, Mw = 197.17 kDa) was obtained from Eisie Chemical Co., Ltd. (Taizhou, China). Hyaluronic acid (HA, MW < 10 KDa) was purchased from Freda (Jinan, China). All other reagents used were of analytical grade.

### 2.2. Methods

#### 2.2.1. Synthesis of Carbon Dots (CD_C-H_)

These CD_C-H_ were synthesized using a one-step hydrothermal method [1]. Typically, HA (0.2 g) and CMCS (0.2 g) are dissolved in 20 mL of ultrapure water and stirred slowly until dissolved. Then, 5 mg of ethylenediamine (EA) added to the mixture was sufficiently mixed and stirred for 30 min. This solution was transferred to a 50 mL Teflon-lined autoclave and heated to 170 °C and kept for a period of 12 h. The reaction products were cooled to room temperature and were filtered through 0.22 μm membrane filters. Then, the resulting solution was dialyzed against a dialysis membrane (molecular weight cut off: 1000 Da) for 24 h to remove free small molecules and redundant precursors. The final product in the dialysis bag was a brown solution, which was then freeze-dried to obtain a brown solid. Finally, CD_C-H_ was obtained and stored in a refrigerator for further use.

#### 2.2.2. Preparation of DOX–CD_C-H_ Complexes

For doxorubicin-bound CD_C-H_ (DOX–CD_C-H_), 3 mL of CD_C-H_ solution (3 mg/mL) was added dropwise to 1 mL of doxorubicin hydrochloride solution (400 μg/mL) under agitation to form a final 4 mL solution system [34,35,36]. This solution was then reacted at room temperature in the dark for 24 h agitations at 300 rpm. After the reaction, in order to remove the solvent in the reaction system, the resultant solution was dialyzed using a dialysis membrane (molecular weight cut off: 1000 Da) against 40 mL deionized water for 2 h. The DOX–CD_C-H_ complexes were obtained after freeze-drying for further characterization.

#### 2.2.3. Characterization 

The spectral properties of synthesized CD_C-H_ were obtained using UV–Vis absorption measurements 6100A (Xiyuan, Shanghai, China) from 200 to 600 nm. The fluorescence spectra were investigated by using a Spectra Max M3 microplate reader (Molecular Devices, San Jose, USA). The size of the nanoparticles and three-dimensional topography of CD_C-H_ were evaluated using a Tecnai G2 F20 high-resolution TEM (HR-TEM), (Thermo Fisher Scientific, Hillsboro, USA) analysis and Bruker Dimension Icon atomic force microscopic (AFM) (Bruker, Karlsruhe, Germany), respectively. FT-IR experiments were performed using a Bruker IFS55 spectrometer (Bruker, Karlsruhe, Germany) using KBr pellets. Crystallinity of CD_C-H_ was studied by the X-ray diffraction (XRD) Brooke D8 ADVANCE III 400 X-ray diffractometer (Bruker, Karlsruhe, Germany) under a voltage of 35 kV and a current of 40 mA with a scanning angular scope of 3–50° (2θ) in step width of 1° per minute. Thermal stability analysis of CD_C-H_ was analyzed by the STA 449 F3 thermogravimetric analyzer (Mettler Toledo, Selb, Germany) of Mettler Toledo. The samples were measured in a nitrogen atmosphere at a temperature range of 25 to 300 °C with a heating rate of 10 °C per min. X-ray photoelectron spectroscopy (ESCALAB 250Xi) (Thermo Scientific, Waltham, MA, USA) was adopted for the chemical compositions and chemical states of CD_C-H_. The hydrate particle sizes and zeta potential of the products were studied with a Malvern Zeta sizer instrument (Melvin, UK).

#### 2.2.4. Fluorescence Quantum Yields

The fluorescence quantum yield (Փ) of the CD_C-H_ was evaluated by a comparative method. In this method, quinine sulfate (literature quantum yield: 54%, dissolved in 0.1 M H_2_SO_4_ and the refractive index of 0.1 M H_2_SO_4_ is 1.33) [9] was selected as reference substance and diluted with 0.1 M H_2_SO_4_ to make its ultraviolet–visible (UV–Vis) absorption (absorption wavelength is 360 nm) less than 0.1. The CD_C-H_ was dissolved in distilled water (the refractive index of distilled water is 1.33) and diluted to colorless so that its UV–Vis absorbance at 360 nm was less than 0.1. 

#### 2.2.5. Buffering Ability

To determine the buffering ability of CD_C-H_ and DOX–CD_C-H_, acid–base titration was performed at a pH range of 12.0–2.0. The buffering property may enable CD_C-H_/DOX–CD_C-H_ to escape from the endosome. In this experiment, a 0.1 M sodium chloride solution was used as control. First, a solution of 1 M sodium hydroxide was used to accurately adjust the pH value to 12.0 for each experimental group. A solution of 0.1 M hydrochloric acid was then added as a titration solution, and the changes in pH of each sample with the titration volume were recorded during the titration process.

#### 2.2.6. Drug Release Studies In Vitro

In vitro drug release experiments were carried out by the USP Apparatus 2 setup using the dynamic dialysis method. The preparations were placed into the dialysis bag with a molecular weight cut off 8000–14,000 Da and were performed at 37 ± 0.5 °C in 50 mL PBS of release medium with different pH values (5.0, 6.8 and 7.4) with a paddle speed of 100 rpm. Samples of 1 mL were withdrawn at predetermined time intervals (1, 2, 4, 6, 8, 10, 12 and 24 h) and immediately replaced with an equal volume of fresh release medium to ensure constant volumes. The fluorescence intensity of samples was measured using a Spectra Max M3 microplate reader (Ex/Em: 495/590 nm).

In order to further explain the drug release mechanism of DOX–CD_C-H_, the kinetic models (Zero-order, First-order, Higuchi and Langmuir kinetic models) were used to fit the release curves.

#### 2.2.7. In Vitro Cytotoxicity Assay

The in vitro cytotoxicity of DOX–CD_C-H_ complexes against NIH-3T3, MCF-7 and 4T1 cells was investigated using the MTT method [37]. The detailed operation process can be summarized as follows: firstly, cells were seeded in 96-well plates (100 μL/well) at a density of 1~2 × 10^4^ cells/well and incubated in a 5% CO_2_ at a temperature of 37 °C for 24 h. Subsequently, the medium was removed and samples at different concentrations were replaced and then again incubated for 24 and 48 h. After incubation, 10 μL of MTT (5 mg/mL) solution was added to each well and incubated at 37 °C for an additional 4 h. After removing that culture medium, 200 μL of DMSO was added to fully dissolve the crystal of formazan, and the absorbance of each well was estimated at 490 nm using Spectra Max M3 microplate reader ((Molecular Devices, San Jose, CA, USA).

#### 2.2.8. Laser Scanning Confocal Microscopy Studies

In order to study the distribution and intracellular uptake of DOX–CD_C-H_ in cells, the NIH-3T3 and 4T1 cells with DOX–CD_C-H_ were analyzed by a laser scanning confocal microscope (LSCM). The cells were incubated in a 24-well plate for 24 h with 105 cells per well. After that, cells treated with DOX–CD_C-H_ samples (100 μg/mL) were incubated for different times (1, 2 and 4 h). One group of cells was treated with free HA solution (MW < 10 KDa, 10 mg/mL) and incubated for 1 h to inhibit the CD_44_ receptor on the cell membrane surface. Then, the medium containing free HA in the well was discarded, and 100 μg/mL of DOX–CD_C-H_ solution was added and incubated for another 6 h. The old medium of all groups was discarded, and the cells were washed three times. Subsequently, the cells were fixed at room temperature for 20 min with 4% paraformaldehyde, followed by 20 min of DAPI staining of the cell nuclei. Finally, the intracellular localization of samples was observed with the LSCM under the same conditions.

#### 2.2.9. In Vivo Antitumor Activity

The animal studies were approved by the Animal Ethical Committee of Shenyang Pharmaceutical University (Shenyang, China) and were conducted according to the guidelines of the Experimental Animal Center. BALB/c mice (6–8 weeks old, Liaoning Changsheng biotechnology Co. Ltd., Benxi, China) as 4T1 tumor-bearing mice models were established by inoculating 4T1 cells suspension into the axillae of mice. When the tumor volume reached approximately 100 mm^3^, the tumor-bearing mice were randomly divided into 4 groups (5 mice/group) and were injected intravenously with CD_C-H_, DOX and DOX–CD_C-H_ complexes at a dose of 5 mg DOX/kg on days 1, 3, and 5, using normal saline as a control. The tumor volume and body weight of the mice were monitored at one-day intervals throughout the 13 days of treatment. At the end of the study, the mice were sacralized, and the tumors were collected to measure their weight. Then, hematoxylin and eosin (H&E) staining was carried out to analyze the histology of tumor tissue and major organs. In addition, terminal deoxynucleotidyl transferase dUTP nick end labeling (TUNEL) was used to analyze apoptosis of tumor cells. 

#### 2.2.10. Statistical Analysis

The results of all measurements were represented as mean ± standard deviation (SD). Data were analyzed via the one-way analysis of variance (ANOVA) by SPASS 26, and differences were expressed as statistically significant with * (*p* < 0.05), ** (*p* < 0.01) and *** (*p* < 0.001).

## 3. Results

### 3.1. Results and Discussion

#### 3.1.1. Preparation and Characterization of CD_C-H_ and DOX–CD_C-H_

HA can be used as a carbon source and carbonized with H_2_SO_4_ to prepare CD_C-H_ [30]. The content of HA as a function targeting on the surface of CD_C-H_ can be adjusted adiabatically by controlling the carbonation time to obtain targeted nanocarriers. Certainly, HA can also be accompanied with glycine, citric acid and polyethyleneimine to prepare carbon dots [28,29,38]. In the preparation of CD_C-H_, the carboxyl group of HA is first dehydrated and condensed with an amino group or hydroxyl group to form an intermediate polymer, which is then the intermediate polymer that is carbonized to produce carbon dots with particle sizes less than 100 nm. In this paper, a one-step hydrothermal method was used to synthesis CD_C-H_ as shown in Figure 1. In preparation, CMCS and HA were selected as carbon sources and EA as a passivation agent to form CD_C-H_. The effect of CMCS and HA with different mass ratios on the fluorescence quantum yield of CD_C-H_ was studied, and the results are shown in Table 1. As the mass of CMCS increases, the fluorescence quantum yield of CD_C-H_ initially increases and then decreases. When the mass ratio of CMCS to HA was 2:2, the fluorescence quantum yield was 11.64%. The excitation and emission spectra of CD_C-H_ at different CMCS to HA mass ratios are shown in Appendix A. Finally, the optimal mass ratio of 2:2 (CMCS: HA: EA, 200/200/5, mg) was selected as the drug carrier.

The optical properties of CD_C-H_ are evaluated as shown in Figure 1. The yellow aqueous solution of CD_C-H_ exhibited strong blue emissions under UV irradiation (365 nm), as shown in Figure 1a (inset). The UV–Vis spectrum of CD_C-H_ exhibited an intense peak at the wavelength of 225 nm and a broad absorption between 250 and 450 nm, which were assigned to conjugated π-π* and n-π* bond structures [39]. Figure 1a shows the photoluminescence spectra of the CD_C-H_ aqueous solution with the maximum excitation and emission wavelengths fixed at 357 and 467 nm. The capability of fluorescence emission spectra of CD_C-H_ aqueous solution was recorded at different excitation wavelengths (330–420 nm, Figure 1b). The CD_C-H_ aqueous solution presents a maximum emission at the excitation wavelength between 340 and 400 nm. In addition, the effect of different pH and dissolution media on the fluorescence emission spectra of CD_C-H_ is also exhibited in Figure 1c,d. Figure 1c shows that the fluorescence intensity of CD_C-H_ decreases when the pH of the solution is lower than 4 or higher than 9. This could be caused by changes in surface morphology caused by the destruction of hydroxyl or amino groups on the surface of the CD_C-H_ by strong acid and strong alkaline. In Figure 1d, the fluorescence spectra and fluorescence intensity of CD_C-H_ in all of the solutions (water, 7.4 PBS, normal saline, DMEM medium and Plasma) were mostly the same, which indicates that the fluorescence properties of the carbon dots are not affected by the dispersion medium. 

The morphological characteristics of CD_C-H_ were characterized by high resolution TEM (HRTEM) and AFM imaging techniques. Figure 2a exhibits that the CD_C-H_ is a spherical morphology with an average statistical diameter of approximately 6 nm (Figure 2b) and has good dispersibility. The HRTEM image (Figure 2c) exhibited a crystalline structure with a lattice distance value of 0.20 nm, which corresponds to the (100) basal spacing of graphitic carbon, suggesting that CD_C-H_ has a graphite-like structure [5]. AFM height images (Figure 2d–f) indicate that the CD_C-H_ present a spherical shape and an average height of 4–5 nm, which also confirmed the TEM characterization. The AFM images in Figure 5a–c exhibit that the average height of DOX–CD_C-H_ was approximately 20 nm [30].

In order to further study the elemental composition and structure of CD_C-H_, X-ray electron spectroscopy was performed to characterize the CD_C-H_. As shown in Figure 3a, the peaks at 284.8, 399.1, and 532.4 eV in the full XPS spectrum revealed the C 1s, N 1s, and O 1s, and their atomic ratios were 61.69%, 13.25% and 25.05%, respectively. The C 1s spectrum shows four peaks (Figure 3b) at 284.8, 285.8, 286.6, and 287.9 eV, which were assigned to C-C, C-N, C-O, and C=O groups of CD_C-H_, individually. The XPS spectrum of N 1s (Figure 3c) shows two distinct peaks at 398.9 and 400.6 eV, which correspond to the C-N and N-H groups. The XPS spectrum of O 1s can be deconvoluted into two peaks at 530.8 and 532.5 eV, which were assigned to the C=O and C-O groups.

The XRD pattern (Figure 4a) suggested that the CD_C-H_ has a broad diffraction peak at 2θ = 21.5°, suggesting that CD_C-H_ is an amorphous structure [13,35]. Thermal stability of CD_C-H_ was analyzed by the STA 449 F3 thermogravimetric analyzer. As shown in Figure 4b, thermal decomposition curves of HA, CMCS and CD_C-H_ have some mass loss that occurs below 100 °C, which is caused by a small amount of moisture in the samples. As can be seen from the carbon residue ratio of the three samples, the decomposition of HA in the molecular chain occurred in the range of 200~250 °C and CMCS is 80% decomposed at 230 °C. The TGA curve proves that the decomposition of CD_C-H_ occurred at 300 °C. This indirectly confirms the synthesis of the target product and the good thermal stability of the CD_C-H_.

Figure 4c shows the FT-IR spectra of the CD_C-H_, DOX and DOX–CD_C-H_ complexes. The broad characteristic peaks from 3000 to 3690 cm ^−1^ are attributed to the stretching vibration of the N-H and O-H groups [40]. The peaks observed at 2923 cm ^−1^ correspond to C-H stretching vibrations. The peaks at 1632, 1402 and 1066 cm^−1^ contributed to the vibrations of C=O, C-N and C-O-C, demonstrating the existence of amido and ester bonds [29]. These groups show that CD_C-H_ have good hydrophilic quality and can absorb DOX through electrostatic interactions. The FT-IR and XPS results show the -C=O, O-H and C–O–C groups in the CD_C-H_, which are specific functional groups of HA. This means that the surface of CD_C-H_ still retains the structural units of HA, which have a strong affinity for the CD_44_ receptor overexpressing on the surface of cancer cells, offering the possibility of further efficient targeted drug delivery. The typical absorption peaks of DOX centered at approximately 2923, 1729, 1582, 1525 and 1413 cm^−1^ all appeared on the DOX–CD_C-H_, which further proved the formation of DOX–CD_C-H_ complexes. As shown in Figure 4d, the UV–Vis spectrum of CD_C-H_ exhibited an intense peak at the wavelength of 225 nm and a broad absorption between 250 and 450 nm, and the UV–Vis spectrum of DOX exhibited an intense peak at 480 nm. However, the DOX–CD_C-H_ exhibited a broad absorption at 350 and 500 nm. There was a small red shift phenomenon, indicating an electrostatic interaction between the CD_C-H_ and DOX. 

Furthermore, the surface charges of CD_C-H_, DOX, and DOX–CD_C-H_ complexes were evaluated via measurement of zeta potential (ζ). As shown in Figure 5d, the zeta potential of CD_C-H_ was −37.3 mV and the DOX was +7.1 mV, suggesting that electrostatic interaction can be formed between the CD_C-H_ and DOX, and the resulting DOX–CD_C-H_ complexes have a zeta potential of −21.3 mV [35]. Table 2 showed that the hydrate particle sizes of DOX–CD_C-H_ were about 201 nm, and the value of PDI was about 0.23.

#### 3.1.2. Buffering Ability of CD_C-H_ and DOX–CD_C-H_

In order to test the pH buffering ability of CD_C-H_ and DOX–CD_C-H_, acid–base titration was used in this study. As shown in Figure 5e, the amount of HCl required to adjust the pH value from 12.0 to 2.0 for each experimental group (normal saline, CD_C-H_ and CD_C-H_ complexes) was recorded. In the titration curve of CD_C-H_ and DOX–CD_C-H_, taking pH value 10 as the node, pH value fluctuated with the change rate of adding HCl, which slowed down the change in pH value to a certain extent. According to the structure, CD_C-H_ can play a role in pH buffering to some extent due to the amino and carboxyl groups on its surface. Therefore, it can be concluded that CD_C-H_ possess acid–base buffering capabilities, which are also the reasons why their fluorescence intensities are stable across different pH media.

#### 3.1.3. Drug-Loading and Release

Drug loading efficiency (LE) and loading content (LC) of DOX–CD_C-H_ were calculated according to the fluorescence intensity calibration curve of DOX, and the results are shown in Appendix A. The LE is about 37%, and the LC is about 6%. The possible mechanism of DOX-loading is electrostatic interaction, which can be formed between the CD_C-H_ and DOX, which was demonstrated by the zeta potential experiment.

The in vitro release profiles of DOX from DOX–CD_C-H_ complexes are illustrated in Figure 5d. There was a relatively high quantity of DOX release from DOX–CD_C-H_ complexes at pH 5.0 (*f*_2_ < 50, compared with the pH medium of 6.8 and 7.4), which showed pH-dependent release behavior. This may be attributed to the strong electrostatic interaction between the drug and the CD_C-H_ carrier. When the system becomes acidic, the charge balance between the drug and the CD_C-H_ carrier will be destroyed, leading to faster drug release. This result exhibits that the DOX–CD_C-H_ complex is highly advantageous in terms of promoting appropriate carriers for cancer drug delivery, as the pH environment of cancerous cells is approximately pH 5.0.

In order to further explain the drug release mechanism of DOX–CD_C-H_, kinetic models were used to fit the release curves. The results are shown in Table 2. The fitting results showed that the release followed the Langmuir kinetic equation (R^2^ > 0.90), indicating drug release induced by a desorption process of physical adsorption. This further indicates that the binding modes of DOX and CD_C-H_ are electrostatically binding.

#### 3.1.4. Cell Viability

The cell viability of CD_C-H_ against NIH-3T3, MCF-7 and 4T1 cells was investigated using the MTT method. As shown in Appendix A, the CD_C-H_ were non-cytotoxic to NIH-3T3 cells, even during incubation over 48 h at concentrations of 12.5–200 μg/mL. However, in Appendix A, a differential toxicity of CD_C-H_ to MCF-7 and 4T1 cells was observed. At a low concentration (12.5–50 μg/mL), the cell viability of the CD_C-H_ was above 75% after incubation with MCF-7 and 4T1 for 24 and 48 h. When the concentration reached 100–200 μg/mL, 48 h cell viability of 4T1 cells dropped to less than 50%, and the cytotoxicity was significantly higher than that of MCF-7. This is consistent with the reported peroxidase activity of CD_C-H_, which can induce apoptosis in tumor cells. In addition, the CD_44_ receptors on the surface of 4T1 cells were more than those of MCF-7 and NIH-3T3; thus, CD_C-H_ is more toxic to 4T1 than MCF-7 and NIH-3T3, confirming that CD_C-H_ is suitable for targeted antitumor applications. In Figure 6, it can be concluded that DOX and DOX–CD_C-H_ show low cytotoxicity to NIH-3T3 cells, and that the cytotoxicity of free DOX is greater than that of the DOX–CD_C-H_, which could be due to the slow release of DOX from the DOX–CD_C-H_ complex. The DOX–CD_C-H_ complexes could reduce the cytotoxicity of free DOX and this result can also be clearly observed in MCF-7 and 4T1 cells (Figure 6b,c). As the concentration increases, DOX accumulates in these cells and shows enhanced cytotoxicity. However, the survival rates of 4T1 cells at all concentrations in the DOX–CD_C-H_ group were significantly lower than in the NIH-3T3 and MCF-7 cells, indirectly confirming CD_C-H_ targeting.

#### 3.1.5. Cell Imaging and Cellular Uptake

To investigate the potential of CD_C-H_ for cell-imaging application and the cellular uptake of DOX–CD_C-H_ in different cells, confocal laser scanning microscopy (CLSM) was applied to monitor DOX–CD_C-H_ complexes. Figure 7 shows the fluorescence images of NIH-3T3 and 4T1 cells that were incubated with DOX–CD_C-H_ (the concentration of DOX was 5 μg/mL) for 2, 4 and 6 h. As shown in Figure 7, the CD_C-H_ present blue fluorescence and are mainly concentrated around the cell membrane and cytoplasm. The DOX–CD_C-H_ complex is localized in both the cell membrane and cytoplasm, and after 6 h, it showed nuclear localization. This could be due to the slow release of the DOX from the DOX–CD_C-H_ complex, and then reaching the nucleus. When the NIH-3T3 and 4T1 cells were cultured with the DOX–CD_C-H_ complexes (in Figure 7 and Figure 8), the fluorescence signal of 4T1 cells was the strongest, indicating greater uptake of DOX–CD_C-H_ complexes by 4T1 cells than that by NIH-3T3 cells. These results are consistent with the cell viability results that show that the DOX–CD_C-H_ complex is more cytotoxic to 4T1 than NIH-3T3 cells. The CLSM results further demonstrate the efficient targeting ability of DOX–CD_C-H_ on CD_44_ tumor cells with excess expression of 4T1. Moreover, to further demonstrate the targeting ability of DOX–CD_C-H_, free HA was added and incubated with 4T1 cells to inhibit the CD_44_ receptor. As shown in Figure 8, the fluorescence intensity was sharply reduced compared to 4T1 cells without inhibitor pretreatment, indicating that the cellular uptake of DOX–CD_C-H_ is mediated mainly by CD_44_ receptors. Therefore, CD_C-H_ are suitable for targeting nanocarriers for tumor cells and have attractive imaging properties.

#### 3.1.6. In Vivo Antitumor Therapy

In order to assess the in vivo therapeutic efficacy of DOX–CD_C-H_, tumor volume and body weight of 4T1 tumor-bearing mice were simultaneously monitored. As shown in Figure 9, DOX and DOX–CD_C-H_ exhibited substantial suppressed tumor growth in comparison to the saline group, while the DOX–CD_C-H_ group exhibited higher inhibition activity than DOX. The tumor inhibition rate was also calculated according to tumor weight. This demonstrated that the DOX–CD_C-H_ group had a value of 64.31% ± 9.29%, which was higher than that of the DOX group (40.90% ± 8.18%, *p* < 0.05). The enhanced antitumor activity was most likely due to the targeted delivery of DOX–CD_C-H_ to the tumor site via CD_44_ targeting and the efficient aggregation of CD_C-H_ nanoparticles for passive targeting, in agreement with the previous results from MTT. Chemotherapy drugs such as DOX are highly toxic and cause weight loss. As shown in Figure 9b, the weight of the DOX group gradually decreases, while the weight of the DOX–CD_C-H_ group is less affected, indicating that the DOX–CD_C-H_ group is able to reduce the toxicity of DOX to some extent.

To further evaluate the in vivo antitumor therapy of DOX–CD_C-H_, H&E and TUNEL staining were performed in tumor tissues of different treatment groups, and pathological analysis was performed, as shown in Figure 9e. After administration with different formulations, different degrees of cell necrosis were observed on day 13. In particular, tumor cells with almost complete necrosis were found in the DOX–CD_C-H_ group. The superior antitumor ability of DOX–CD_C-H_ was further confirmed by TUNEL staining, which was consistent with H&E staining, with tumor slides from the DOX–CD_C-H_ group producing more green apoptotic regions than those from the other treatment groups.

The safety of DOX–CD_C-H_ was further assessed based on previous in vivo antitumor studies. Historical changes in the major organs were observed by H&E staining. Figure 10 shows that no morphological changes were observed in the major organs of the mice in each group, and no significant necrosis or injury was observed.

### 3.2. Formatting of Mathematical Components

The photo-luminescence (PL) emission spectra of all samples were determined by a fluorescence photometer with an excitation wavelength of 360 nm, and the fluorescence quantum yield was calculated by the following equation (Equation (1)).
(1)$${Փ}_{t}={Փ}_{s}\times \frac{I_{t}}{I_{s}}\times \frac{A_{s}}{A_{t}}\times \frac{\eta_{t}}{\eta_{s}}$$
where *s* and *t* refer to the standard sample and the test sample, respectively; Փ is the fluorescence quantum yield. *I* is the integral area of the fluorescence emission spectrum. A is the UV–Vis absorbance value, and η is the refractive index of the solvent.

The similarity factor f2 method was used to compare the dissolution profiles, and the value of *f*_2_ was calculated using the following equation (Equation (2)).
(2)$$f_{2}=50\times log\left\{{\left[1+\left(\frac{1}{n}\right)\sum {\left(R_{t}-T_{t}\right)}^{2}\right]}^{-0.5}\times 100\right\}$$
where *n* is the number of time points, *R_t_* is the dissolution value of the reference at time t, and *T_t_* is the dissolution value of the test at time *t*. The release profiles were significantly different if *f*_2_ < 50.

Cell viability was calculated using the following equation (Equation (3)).
(3)$$\mathrm{Cell}\;\mathrm{viability}\%=\frac{A_{sample}-A_{blank}}{A_{control}-A_{blank}}\times 100\%$$

The tumor volume was calculated using the following equation (Equation (4)).
V (mm^3^) = 0.5 × length × width × width(4)

## 4. Discussion

Using HA and CMCS as carbon sources in a one-step hydrothermal method, a novel green carbon dot nano-drug delivery system with targeting and cell imaging was developed and characterized. In this process, the use of two polymers (HA and CMCS) and EA to prepare CD_C-H_ by a one-step hydrothermal method was not reported. The synthesis mechanism of CD_C-H_ is that the carboxyl group of HA polymerizes with the amine group of EA or the hydroxyl group of CMCS to form an amide or ester polymer. Based on optical properties, chemical structure, elemental composition and surface state, CD_C-H_ is a 6 nm graphene-like, amorphous, highly dispersed nanoparticle containing doped N atoms and emitting blue fluorescence. The large numbers of carboxyl, hydroxyl, and amino groups on the surface of the CD_C-H_ give it excellent water dispersal and moderate stability, which also provide conditions for their structural stability in different biological environments. In addition, the acid–base buffering ability of CD_C-H_ stabilizes its fluorescence intensity across different pH media, and this buffering ability is due to the large number of carboxyl, hydroxyl and amino groups on their surfaces. Furthermore, FTIR and XPS results show that structural units of HA with strong affinity for CD_44_ receptor expression on tumor cells are still retained on the surfaces of CD_C-H_. This makes it possible to deliver drugs in a targeted manner and reduce toxicity and side effects. In our study, DOX was used as a model drug to form nanocomplexes with negatively charged CD_C-H_ on the surface via electrostatic adsorption. The surface charge value (−21.3 mV) of this complex showed that it had excellent stability, and the in vitro release results showed that it had pH responsive release ability, which provided some basis for its stability in vivo. Most of the synthesis of targeted functional carbon sites is conducted by post-modification. This method is time-consuming and laborious in purification and is not commonly used. Functionalized carbon dots prepared by one-step synthesis are simple, short lived and low cost. However, the surface properties of carbon dots in one-step synthesis methods are difficult when trying to accurately obtain carbon dots by controlling the preparation conditions. Thus, the development of a process that can regulate the process conditions for the preparation of versatile and accurate carbon dots remains a key difficulty for the application of carbon dots.

In this study, we reported the synthesis of novel CD_C-H_ by a simple one-step hydrothermal method using natural molecular materials (HA and CMCS) and their application in the treatment of tumors. In both cytological experiments and in vivo pharmacokinetic studies, we found enhanced tumor therapeutic effects. This is closely related to the hyaluronic acid on the surface of the carbon dot and the particle size of the carbon dot. The targeting ability of HA and the particle size of the carbon dot increase the active and passive targeting ability of HA, which enhances the efficacy. This novel carbon dot nano-drug delivery system fabricated from polymers provides a fresh material choice and idea for functional carriers and tumor therapies.

## 5. Conclusions

In this work, a novel photo-luminescent targeted nanocarrier, CD_C-H_, composed of biocompatible HA and CMCS was synthesized by a one-step hydrothermal method, where DOX is bound with the CD_C-H_ via electrostatic interactions to form the DOX–CD_C-H_ complex as a targeted antitumor drug delivery system. Compared to other synthetic methods, this method has a high quantum yield of 11.64%, is simple, and is easy to scale up. Methods such as UV, PL, TEM, AFM, XPS, TGA, XRD, FT-IR and zeta potential were applied to characterize the characterization of the CD_C-H_ and DOX–CD_C-H_ complexes, and these results confirm the successful preparation of the CD_C-H_ and DOX–CD_C-H_ complexes. Furthermore, the in vitro release study showed that DOX–CD_C-H_ has pH-dependent DOX release behavior in vitro at different pH mediums (5.0, 6.8 and 7.4) for 24 h. Confocal laser scanning microscopy confirmed the efficient imaging and cellular uptake of DOX–CD_C-H_ in tumor cells with overexpressed CD_44_ receptors. Moreover, the results of MTT indicated that the DOX–CD_C-H_ complexes could induce apoptosis in 4T1 cells, reducing the cytotoxicity of free DOX-induced apoptosis. Significantly, in vivo antitumor studies demonstrated that the DOX–CD_C-H_ complex displayed superior therapeutic effects. In summary, the successful development of the excellent biopolymer-based CD_C-H_ may serve as a promising candidate for bio-imaging and target delivery nanocarriers for tumor cells with high CD_44_ receptor expression. 

## Data Availability

Not applicable.

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
