# Peer review of "Easy Synthesis and Characterization of Novel Carbon Dots Using the One-Pot Green Method for Cancer Therapy"

_pharmaceutics, 2022, doi:10.3390/pharmaceutics14112423_

Round 1

Reviewer 1 Report

The manuscript by Wang et al. is aimed at DOX loaded green luminescent carbon dots cancer cells targeting and therapy. Targeting function is due to expected HA affinity to cell membrane receptors. Due to the novelty and originality brought together this manuscript can be of great interest for the readers of Pharmaceutics. However, there are several flaws to be eliminated. Several issues have to be carefully addressed before accept recommendation is given.

1     Does HA remain stable after solvothermal synthetic procedure enough to enable surface coating and targeting function? The authors should know that HA is extremely unstable at high temperatures. This should carefully be considered by the authors.

2     Conjugation requires covalent bonding, isn’t it? For electrostatic deposition statement “For doxorubicin conjugated CDC-H (DOX-CDC-H)” is incorrect.

3     Extensive English language editing is required.

4     Solvothermal conditions should be specified at Scheme 1.

5     Axes numbering and legends on the Figure 1 very poorly readable. Please fix.

6     Very poor signal-to-noise ratio for emission spectra at Figure 1b,c should be improved.

7     The paper lacks of proper literature citation. Following very fresh and topic-relevant papers 10.3390/nano11113080, 10.1016/j.pdpdt.2022.103099, 10.1016/j.carbon.2022.07.065, 10.1002/ppsc.202200017, 10.3390/coatings12010097, 10.1021/acsabm.2c00235 should be cited in introductory part.

8     Please add excitation and emission spectra of the CDC-H at different HA CMCS mass ratios.

9     Figure 2b scale bar is not readable.

10   Please clearly illustrate the line in AFM topography images (Figure 2c and 5a).

11   What is the reason of zeta potential of DOX positive and equal to +7.1 mV. Can ZP be obtained for molecular level dispersed systems?

12   Why DAPI staining is so bad at Figure 7?

13   Why 4T1 cells are not apoptic after 4-6 hours of DOX incubating?

14   Lijie Wang is metioned two times in the author list. Please fix it. 

Reviewer 2 Report

The major aim of this research article is to prepare green-colored carbon dots (CDs) by using hyaluronic acid and carboxymethyl chitosan in the presence of ethylenediamine (via hydrothermal process) for in vitro and in vivo breast cancer therapy. Though the work is extensive and the article is written interestingly, few major changes needed to be done based on following comments before publication.

·         Typographical and language grammar mistakes should be modified. For instance in the title itself, it is given as “a novel green carbon dots”. This should be only “novel green carbon dots”.

·         The authors claim that the CDs exhibit green fluorescence. However, from inset in Fig 1 (a), it can be clearly noted that blue fluorescence is emitted under UV illumination. So, how this can be claimed to be green. Need explanation and corresponding changes throughout the article.

·         In almost all graphs, the X & Y axes, and the inside texts and numbers are not clearly visible. It should be made enlarged and bold. In few graphs, title and/or units in X & Y axes are missing.

·         A better TEM image with more number of CDs can be included for improving the quality of paper.

·         Similar kinds of research articles are already present. So, the need for this article should be clearly presented in the “Introduction” part.

Reviewer 3 Report

In this work, Easy synthesis and characterization of a novel green carbon dots for cancer therapy. The idea of this research is interesting to readers. The background is well studied and the presentation of the method is very clear and sound, but there are some minor issues to be addressed:

Comments

The author should provide some quantitative information in the abstract section.

The author need to provide suitable reference in the section 2.2.2. Preparation of the DOX-CDC-H complexes.

If possible author cite some important following references in the introduction section; (10.1016/j.microc.2021.106280, 10.1039/d1ay01746b, 10.1039/d1ay02077c).

The author should improve the quality of TEM image in Figure 2.

The author should cite important reference in the Results and discussion section.

In Figure 10. H&E staining of major organs after different treatments. This figure legend should provide in briefly.

Round 2

Reviewer 1 Report

Reviewer thanks authors for careful revision. Manuscript was greatly improved and can surely be accepted at this stage. Recommendation accept in the present form.

Reviewer 2 Report

The modified manuscript can be accepted for publication.